# Bandwidth Detection of Graph Signals with a Small Sample Size

**DOI:** 10.3390/s21010146

**Published:** 2020-12-28

**Authors:** Xuan Xie, Hui Feng, Bo Hu

**Affiliations:** Research Center of Smart Networks and Systems, Fudan University, Shanghai 200433, China; xxie15@fudan.edu.cn (X.X.); bohu@fudan.edu.cn (B.H.)

**Keywords:** graph signals, bandwidth detection, bayesian score test

## Abstract

Bandwidth is the crucial knowledge to sampling, reconstruction or estimation of the graph signal (GS). However, it is typically unknown in practice. In this paper, we focus on detecting the bandwidth of bandlimited GS with a small sample size, where the number of spectral components of GS to be tested may greatly exceed the sample size. To control the significance of the result, the detection procedure is implemented by multi-stage testing. In each stage, a Bayesian score test, which introduces a prior to the spectral components, is adopted to face the high dimensional challenge. By setting different priors in each stage, we make the test more powerful against alternatives that have similar bandwidth to the null hypothesis. We prove that the Bayesian score test is locally most powerful in expectation against the alternatives following the given prior. Finally, numerical analysis shows that our method has a good performance in bandwidth detection and is robust to the noise.

## 1. Introduction

Graph signal (GS) is a versatile model for describing information in irregular domains, which has been widely used in sensor networks [1,2], biological networks [3,4] and image and 3-D point cloud processing [5,6]. Graph signal processing (GSP) theory generalizes the classic discrete signal processing theory to irregular domains by introducing graph Fourier transform (GFT) [7,8]. In most real world networks, adjacent vertices usually have similar signals, which leads to bandlimitedness or approximate bandlimitedness in graph spectral domain. For example, the differences of temperature measured by sensors within short distances may not vary a lot; people tend to have similar interests or views with their friends in the social network.

Bandwidth is a widely used prior of GS in selecting the best sensor placements to monitor spatial phenomena [9,10,11], low-pass filter designed for denoising [7,8,12] and estimating the signals on all the sensors from partial observations [13,14]. However, the bandwidth of GS is typically unknown in practice. There are only a few works considering the problem of bandwidth detection. The bandwidth estimation in Reference [9] relies on noise-free samples. The scenario of noisy observations is considered in Reference [15], but it only applies to the spectral sparse case. Meanwhile, the estimation accuracy of the bandwidth in Reference [15] highly depends on the parameter controlling the sparsity of the GS, and the process of finding an optimal parameter is time-consuming.

The detection of bandwidth for a GS aims to detect the number of spectral components in a GS, which can be seen as a model selection problem in linear regression. The well-known model selection procedures like Akaike information criterion (AIC) and Bayesian information criterion (BIC) [16] pick the most likely model under some criteria. But they do not consider the significance of the chosen model. In contrast, hypothesis testing, such as likelihood ratio (LR) test and *F*-test can provide the significance of a model. If the bandwidth of a GS is *k*, the largest index of non-zero elements in frequency coefficients should be *k*. Let the vertex number of a GS be *N*, the regression coefficients of the last N−k frequency components are denoted by ${\hat{f}}_{-k}$. To test whether the bandwidth of the GS is *k*, the hypothesis can be expressed as
(1)$$\begin{array}{c} H_{0}:{\hat{f}}_{-k}=0,\;H_{A}:{\hat{f}}_{-k}\neq 0. \end{array}$$

The null hypothesis will be rejected if the model with the assumed bandwidth *k* is not significant enough. Therefore, the bandwidth of GS can be detected by applying tests over all the assumed bandwidths to see whether there is a model achieving the given significance.

For large-scale GS, such as social networks, it is impossible to sample the signal on every vertex due to huge data collection cost, which means we can only have the sample size M<N−k sometimes. In this high dimensional situation, the conventional test for testing regression coefficients such as LR test and *F*-test is infeasible, which will be explained in Section 3.1. There are plenty of literature on testing the coefficients of high dimensional linear regression. For example, Reference [17,18] propose new test statistics by modifying the *F*-statistic, and Reference [19] introduces Bayesian priors to the parameters being tested. When k≪N, the spectrum of the GS can be seen as sparse. Several approaches reconstruct the signal in compressive sensing which estimate the bandwidth as a by-product [20,21]. The frequency components detected by compressive sensing methods can help estimate the spectrum sparse signal well. However, the bandwidth obtained by the compressive sensing methods may be inaccurate since the choose frequency components in the whole space without considering the bandlimitedness. More accurate bandwidth is need in some applications like filter design and sampling set design of which the performance highly depend on the bandwidth prior.

In this letter, we try to detect the bandwidth of bandlimited GS with small sample size, which is not much larger than the bandwidth and there is no sparsity constraint to the spectrum. The bandwidth is detected by a forward multi-stage test. The bandwidth of model being tested is increasing over stages and the bandwidth is obtained when the null hypothesis is accepted. Since the samples are not adequate, we try to use the limited samples to better distinguish the assumed bandwidth from those close to it in each stage for a small detection bias. Therefore, we customize priors for ${\hat{f}}_{-k}$ in each stage to describe our attention on testing whether some elements in ${\hat{f}}_{-k}$ are non-zero. By doing so, we only use the limited samples to focus on testing a few alternatives in each stage. Bayesian score test [19] is adopted in this paper, but we do not give uniform prior to the parameters being tested as Reference [19] does. Since the bandwidth being tested is increasing over stages, our attentions on coefficients of different frequencies are also updated in each stage which makes our method able to locate the true bandwidth with a small bias even though the samples are insufficient.

The widely used multi-stage test for model selection, stepwise regression ([16] Chapter 10) does poorly with a small sample size since the estimation of the regression coefficients in it is biased [22]. In contrast, our method is proved to be locally most powerful (LMP) in expectation against alternatives following the given prior, which means it performs well in distinguishing among models with neighbouring bandwidth even using a small amount of samples in each stage.

## 2. Problem Formulation

Consider an *N*-vertex undirected connected graph G=(V,E,W), where V is the vertex set, E is the edge set and W is the N×N weighted adjacency matrix with $W\left(i,j\right)=w_{ij}$. A GS defined on V can be represented as a vector $f\in R^{N}$, and its element $f_{i}$ represents the signal value at the *i*-th vertex in V. Laplacian matrix of the undirected graph is given by L=D−W, where D is the diagonal degree matrix $\mathrm{diag}\{d_{1},\cdots ,d_{N}\}$ with $d_{i}=\sum_{j}w_{ij}$. Since $w_{ij}=w_{ji}$ for undirected graphs, Laplacian matrix is symmetric. Therefore, it has real non-negative eigenvalues $0=\lambda_{1}\leq \lambda_{2}\leq \cdots \leq \lambda_{N}$ and an orthogonal set of eigenvectors V. As a result, the spectral decomposition of graph Laplacian is given as $L=V\Lambda V^{T}$, where Λ is a diagonal matrix of eigenvalues. The columns of V denoted by ${\left\{v_{i}\right\}}_{1\leq i\leq N}$ are regarded as the graph Fourier bases and the eigenvalues are regarded as frequencies [7]. The expansion coefficients of f corresponding to eigenvectors are defined as $\hat{f}$.

A GS is called *bandlimited* when there exists a K∈{0,1,⋯,N−1} such that its GFT $\hat{f}$ satisfies ${\hat{f}}_{i}=0$ for all i≥K [10]. The smallest such *K* is called the bandwidth of f. If all the frequency coefficients are non-zero, the GS is not bandlimited. If f is a signal with bandwidth *K*, then it satisfies $f=V_{K}{\hat{f}}_{K}$, where $V_{K}$ denotes the first *K* columns of V and ${\hat{f}}_{K}$ denotes the first *K* coefficients of $\hat{f}$.

Suppose that M(M<N) noisy observations $y\in R^{M}$ are sampled from $f\in R^{N}$ to detect the bandwidth, the observation model can be summarized as
(2)y=Ψ(f+w),
where w is an N×1 vector with *i*th element representing the observation noise on the *i*th vertex and $\Psi :R^{N}\to R^{M}$ is the sampling matrix, of which the element in the *i*th row and *j*th column is defined as
(3)$$\begin{array}{c} {\left(\Psi \right)}_{i,j}=\left\{\begin{array}{cc} 1, & \mathrm{if}\;\mathrm{the}\;j\mathrm{th}\;\mathrm{vertex}\;\mathrm{is}\;\mathrm{the}\;i\mathrm{th}\;\mathrm{sample};\\ 0, & \mathrm{otherwise}. \end{array}\right. \end{array}$$

We use a simple example to illustrate the role of the sampling matrix. For a graph with 5 vertices and signal $f={\left[f_{1},\cdots ,f_{5}\right]}^{T}$, if the 1st, 2nd and 4th vertices are sampled, then the sampling matrix is a matrix with 3 rows and 5 columns as follow
(4)$$\begin{array}{c} \Psi =\left[\begin{array}{ccccc} 1 & 0 & 0 & 0 & 0\\ 0 & 1 & 0 & 0 & 0\\ 0 & 0 & 0 & 1 & 0 \end{array}\right]. \end{array}$$

The sampled graph signal is $\Psi f={\left[f_{1},f_{2},f_{4}\right]}^{T}$.

In this paper, we assume that the observation noise on each sample follows the Gaussian distribution $N(0,\sigma^{2})$ and the noise on all the samples are independently and identically distributed (*i.i.d.*), so the observation follows $w∼N(0,\sigma^{2}I_{N\times N})$, where $I_{N\times N}$ is an N×N identity matrix.

By dividing the columns of V into two parts $V_{k}$ and $V_{-k}$ for any assumed bandwidth k∈{1,2,⋯,N−1}, the observation model (Equation 2) can be rewritten as
(5)$$\begin{array}{c} y=U_{k}{\hat{f}}_{k}+U_{-k}{\hat{f}}_{-k}+\Psi w, \end{array}$$
where ${\hat{f}}_{k}$ and ${\hat{f}}_{-k}$ denote the first *k* and the last N−k elements of $\hat{f}$, respectively, and $U_{k}=\Psi V_{k}$, $U_{-k}=\Psi V_{-k}$. If the actual bandwidth K=k, $H_{0}$ in (Equation 1) will be accepted. Otherwise, $H_{0}$ will be rejected.

For a GS with actual bandwidth *K*, the hypothesis with assumed bandwidth k<K should all be rejected. Therefore, we detect the bandwidth of GS in a forward multi-stage way with assumed bandwidth increasing over stages. The bandwidth is obtained when the null hypothesis is accepted. By doing so, we reduce the number of models to be tested from *N* to *K*.

## 3. Bandwidth Detection

### 3.1. High Dimensional Challenge

*F*-test and LR test are common approaches to test the hypothesis (Equation 1) in linear regression model. Let the maximum likelihood estimation (MLE) of $\hat{f}$ under $H_{0}$ and $H_{1}$ be ${\hat{f}}_{R}$ and ${\hat{f}}_{U}$, respectively. The corresponding sum of squared residuals (SSR) are denoted by $\mathrm{SSR}({\hat{f}}_{R})$ and $\mathrm{SSR}({\hat{f}}_{U})$. Then, LR test statistic ([23] Chapter 10) equals $M\log(\mathrm{SSR}\left({\hat{f}}_{U}\right)/\mathrm{SSR}\left({\hat{f}}_{R}\right))$, which is asymptotically distributed as $\chi^{2}\left(N-k\right)$ as the sample size approaches to infinity. When the sample size is small, the test statistic of LR test has no specific distribution which makes it hard to implement. *F*-test statistic ([23] Chapter 10) equals
(6)$$\begin{array}{cc} \frac{\mathrm{SSR}\left({\hat{f}}_{U}\right)-\mathrm{SSR}\left({\hat{f}}_{R}\right)/N-k}{\mathrm{SSR}({\hat{f}}_{R})/M-k}\; & =\frac{y^{T}M_{k}U_{-k}{\left(U_{-k}^{T}M_{k}U_{-k}\right)}^{-1}U_{-k}^{T}M_{k}y/N-k}{y^{T}M_{k}\left(I_{M\times M}-U_{-k}{\left(U_{-k}^{T}M_{k}U_{-k}\right)}^{-1}U_{-k}^{T}\right)M_{k}y/M-k}, \end{array}$$ where $M_{k}=I_{M\times M}-U_{k}{\left(U_{k}^{T}U_{k}\right)}^{-1}U_{k}^{T}$ is a projection matrix that projects signals to the complementary subspace of the column space of $U_{k}$. *F*-test is exact in finite samples, however, to ensure that $U_{k}^{T}U_{k}$ in $M_{k}$ is invertible, we have to let M≥k. Similarly, to ensure that $U_{-k}^{T}M_{k}U_{-k}$ is invertible, there must be M≥N−k. Therefore, (Equation 6) requires that M≥max(K,N−k), which means the sample size needs to be bigger than the assumed bandwidth, as well as the dimension of ${\hat{f}}_{-k}$. For large-scale GS, the sampling cost for bandwidth detection using *F*-test or LR test is too high. Therefore, a test with small sample size is preferred. The sample size only needs to be larger than the assumed bandwidth, that is, M>k, in score test ([23] Chapter 10). However, when N−k>M, there may exist some alternatives which have ${\hat{f}}_{-k}\neq 0$ but $U_{-k}{\hat{f}}_{-k}=0$. We can never hope to have any power against these alternatives. Since the GS is known to be bandlimited, we should pay more attention on testing whether the low frequency coefficients in ${\hat{f}}_{-k}$ are non-zero than the high frequency coefficients. Thus, we adopt Bayesian score test [19] to allocate our attention among the alternatives by designing a prior for ${\hat{f}}_{-k}$. Different from Reference [19], which pays equal attention to all the alternatives, we customize a prior for ${\hat{f}}_{-k}$ to fit our bandwidth detection purpose.

### 3.2. Design of the Prior

In the test for each bandwidth, the alternative that ${\hat{f}}_{-k}=\alpha$ and ${\hat{f}}_{-k}=-\alpha$ for every α≠0 contributes equally in rejecting $H_{0}$, so we assign $E({\hat{f}}_{-k})=0$ to make the prior unbiased, where E(·) denotes the expectation of the given stochastic variable. The covariance matrix of ${\hat{f}}_{-k}$ can be assigned as
(7)$$\begin{array}{c} E\left({\hat{f}}_{-k}{\hat{f}}_{-k}^{T}\right)=\tau^{2}\Sigma_{-k}, \end{array}$$
where $\Sigma_{-k}$ is a positive semidefinite k×k matrix to be designed and $\tau^{2}$ is an unknown parameter. In this paper, we let $\Sigma_{-k}$ be a diagonal matrix, then each of its diagonal element is the variance of the corresponding element in ${\hat{f}}_{-k}$. Since $E({\hat{f}}_{-k})=0$, a larger variance indicates the corresponding element in ${\hat{f}}_{-k}$ is more likely to have a larger amplitude. To decide whether $H_{0}$ should be accepted, more attention should be payed to the element in ${\hat{f}}_{-k}$ with larger variance. Thus, *the attention among elements in ${\hat{f}}_{-k}$ in bandwidth detection is linked to their variances in the prior*. The attention is allocated by designing a $\Sigma_{-k}$.

In our forward multi-stage test, the bandwidth being tested updates over stages, thus $\Sigma_{-k}$ also needs to be updated. To avoid the the multi-stage test accepting the null hypothesis too early, the test should be more distinguishable from the model with bandwidth close to the null hypothesis in each stage. Therefore, we use the limited samples to make the test concentrate on distinguishing bandwidth close to *k*. The design of $\Sigma_{-k}$ follows the guideline that as the frequency increases, element in ${\hat{f}}_{-k}$ with higher frequency is paid less attention. Since we cannot determine a bandwidth larger than M−1 with sample size *M*, we set the attention on ${\hat{f}}_{-k}$ with frequency larger than *M* to a small constant δ. For example, the *i*th diagonal element of $\Sigma_{-k}$ can be designed to be
(8)$$\begin{array}{c} {\left(\Sigma_{-k}\right)}_{i,i}=\frac{1}{Z}\exp\left(-\frac{i^{2}}{2\sigma^{2}}\right)+\delta , \end{array}$$
where $Z=\exp\left(-1/(2\sigma^{2})\right)$ is the normalization factor. According to the three-standard-deviations property of normal distribution, σ is set to be (M−k)/3 to make the attention on coefficients with frequencies larger than *M* equals to δ with high probability. Equation (Equation 8) is not the only choice of $\Sigma_{-k}$, but this form is applicable. The details about how it affects the power of the test is analyzed later in Section 3.3.

As the multi-stage test moves forward, the attentions among frequency coefficients update, and in each stage, the test is distinguishable among bandwidth close to *k*, which makes the small-bias bandwidth detection with small sample size possible. An illustration of the attention update is shown in Figure 1.

We can complete the specification of the distribution of ${\hat{f}}_{-k}$ by choosing a value for $\tau^{2}$ and a distribution shape. Let $L_{k}\left({\hat{f}}_{-k};y\right)$ be the likelihood of ${\hat{f}}_{-k}$, for a specific prior distribution of ${\hat{f}}_{-k}$, the likelihood of $\tau^{2}$ is
(9)$$\begin{array}{c} {\bar{L}}_{k}\left(\tau^{2};y\right)=\int L_{k}\left({\hat{f}}_{-k};y\right)\pi \left({\hat{f}}_{-k}|\tau^{2}\right)d{\hat{f}}_{-k}, \end{array}$$
where $\pi ({\hat{f}}_{-k}|\tau^{2})$ is the distribution of ${\hat{f}}_{-k}$ for a given $\tau^{2}$.

Given ${\bar{L}}_{k}\left(\tau^{2};y\right)$, we convert hypothesis (Equation 1) to
(10)$$\begin{array}{c} {\bar{H}}_{0}:\tau^{2}=0,\;{\bar{H}}_{A}:\tau^{2}\neq 0. \end{array}$$

$\tau^{2}=0$ implies ${\hat{f}}_{-k}=0$, since they lead to the same distribution of y. Thus $H_{0}:{\hat{f}}_{-k}=0$ will be rejected if ${\bar{H}}_{0}$ is rejected. *The score test of ${\bar{H}}_{0}$ is named the Bayesian score test of $H_{0}$ with the given prior of ${\hat{f}}_{-k}$.*

The score test of ${\bar{H}}_{0}$ in stage *k* is a one-sided test for one parameter, the test statistic is
(11)$$\begin{array}{cc} S_{\Sigma_{-k}} & =\frac{d}{d\tau^{2}}\log{\bar{L}}_{k}\left(0;y\right), \end{array}$$
(12)$$\begin{array}{cc} \; & =\frac{1}{2}g_{k}^{T}\Sigma_{-k}g_{k}-\frac{1}{2}\mathrm{tr}\left(\Sigma_{-k}F_{k}\right), \end{array}$$
where $\mathrm{tr}(\Sigma_{-k}F_{k})$ denotes the trace of $\Sigma_{-k}F_{k}$. ${\bar{H}}_{0}$ will be rejected when $S_{\Sigma_{-k}}\geq t$ for some threshold *t*. From (Equation 11), we can find that the test statistic of $\tau^{2}$ can be seen as the slope of log-likelihood function of $\tau^{2}$ under ${\bar{H}}_{0}$. The slope will equal to 0 when $\tau^{2}$ equals its MLE ${\hat{\tau }}^{2}$. The more closer ${\hat{\tau }}^{2}$ to 0, the more closer $S_{\Sigma_{-k}}$ to 0. If $S_{\Sigma_{-k}}$ is larger than a given *t*, which means ${\hat{\tau }}^{2}$ is much larger than 0, then ${\bar{H}}_{0}$ will be rejected. Equation () is given by Reference [19], where $g_{k}=\sigma^{-2}U_{-k}M_{k}y$ and $F_{k}=\sigma^{-2}U^{T}U$ are the gradient and Fisher information matrix of $\log L_{k}\left(0;y\right)$, respectively.

### 3.3. Method

Considering that the second part of () is not related to the observations, it is more convenient to work with the equivalent test statistic $\sigma^{-2}y^{T}M_{k}U_{-k}\Sigma_{-k}U_{-k}^{T}M_{k}y$. Because $\sigma^{2}$ is not known, we plug in the MLE result $\hat{\sigma^{2}}\propto y^{T}M_{k}y$ under the null hypothesis, the resulting test statistic is
(13)$$\begin{array}{c} S_{\Sigma_{-k}}=\frac{y^{T}M_{k}U_{-k}\Sigma_{-k}U_{-k}^{T}M_{k}y}{y^{T}M_{k}y}. \end{array}$$

*Interpretation of the test statistic*: $M_{k}y$ indicates the part of y out of the range of $U_{k}$. The numerator of (Equation 13) can be rewritten as $Q=\sum_{i=1}^{M}{\left(\Sigma_{-k}\right)}_{i,i}Q_{i}$ with $Q_{i}={\left(y^{T}M_{k}U_{k+i}\right)}^{2}$. Larger $Q_{i}$ indicates a larger energy of $M_{k}y$ lying in the range of $U_{k+i}$, which links to a larger amplitude of the *i*th element in ${\hat{f}}_{-k}$. $S_{\Sigma_{-k}}$ equals *Q*, which is a weighted sum of $Q_{i}$, normalized by the energy of $M_{k}y$. When the weight ${\left(\Sigma_{-k}\right)}_{i,i}$ decreases with *i*, $S_{\Sigma_{-k}}$ will be larger if $Q_{i}$ is larger for smaller *i*. Therefore, the test is more powerful against $H_{A}$ that has non-zero frequency coefficients in low frequency than that has non-zero frequency coefficients in high frequency. This is in accordance with the purpose of our design that the test should be more distinguishable from the model with bandwidth close to *k*.

Finally, we decide whether the null hypothesis is accepted by a *p*-value threshold. For a given significance level α, if p≤α, the null hypothesis is rejected. Let $S_{-k}$ be the observation value of $S_{\Sigma_{-k}}$, then the *p*-value of the test is
(14)$$\begin{array}{cc} p & =P_{H_{0}}\left(S_{\Sigma_{-k}}>S_{-k}\right),\\ \; & =P_{H_{0}}\left(y^{T}M_{k}\left(U_{-k}\Sigma_{-k}U_{-k}^{T}-S_{-k}\right)M_{k}y>0\right). \end{array}$$

Since $M_{k}y=M_{k}\left(y-U_{k}{\hat{f}}_{k}\right)=M_{k}w$ under null hypothesis, it follows a multivariate normal distribution. Thus, *p* is the probability that the quadratic form of normal variables is non-negative. The distribution of the quadratic forms in normal variables is approximated by a $\chi^{2}$ distribution in Reference [24] and *p*-value can be calculated approximately as,
(15)$$\begin{array}{c} p=\left\{\begin{array}{cc} P(\chi_{d_{k}}^{2}>d_{k}-h_{k}), & \;\mathrm{tr}(X_{k}^{3})>0;\\ P(\chi_{d_{k}}^{2}<d_{k}-h_{k}), & \;\mathrm{tr}(X_{k}^{3})<0;\\ \Phi \left(\frac{\mathrm{tr}(X_{k})}{\sqrt{2\mathrm{tr}(X_{k}^{2})}}\right), & \;\mathrm{tr}(X_{k}^{3})=0, \end{array}\right. \end{array}$$
where $X_{k}=M_{k}U_{-k}\Sigma_{-k}U_{-k}^{T}M_{k}-S_{-k}M_{k}$, $d_{k}={\left[\mathrm{tr}\left(X_{k}^{2}\right)\right]}^{3}/{\left[\mathrm{tr}\left(X_{k}^{3}\right)\right]}^{2}$, and $h_{k}=\mathrm{tr}\left(X_{k}^{2}\right)\mathrm{tr}\left(X_{k}\right)/$$\mathrm{tr}(X_{k}^{3})$.

Our algorithm for the multi-stage bandwidth detection is shown in Algorithm 1. If p≤α for all the stages, the output of Algorithm 1 will be ‘None’, which means the bandwidth cannot be determined with the given sample size. Then we can say that the bandwidth is larger than M−1.
**Algorithm 1:** Multi-stage Bandwidth Detection.  **Input** Samples y and significance level α;  **Output** Detected bandwidth *K* or None; 1: **for**
k=1,2,⋯,M−1
**do** 2:  Calculate test statistic according to (Equation 13); 3:  Calculate the *p*-value of the test according to (Equation 15); 4:  **if**
p>α
**then** 5:   Let bandwidth K=k and **stop**; 6:  **else** 7:   k=k+1; 8:  **end if** 9: **end for**

### 3.4. Power Analysis

In the situation N−k>M, there may exist some alternatives which have ${\hat{f}}_{-k}\neq 0$ but $U_{-k}{\hat{f}}_{-k}=0$. It is impossible to find a test which is optimal against all the alternatives, that is, uniformly most powerful. If the true ${\hat{f}}_{-k}$ has large deviations from $H_{0}$, it is very easy to detect. However, if the deviations are small, the detection becomes harder. In the Bayesian score test of $H_{0}$, the deviations of the true ${\hat{f}}_{-k}$ from $H_{0}$ is denoted by τξ, where ξ is assumed to follow a prior distribution with E(ξ)=0 and $E\left(\xi \xi^{T}\right)\propto \Sigma_{-k}$ and $\tau^{2}$ indicates how large the deviation is. Next, we will analyze the power of the Bayesian score test of $H_{0}$ when $\tau^{2}$ is small.

**Definition 1** (Locally most powerful (LMP) [25]). *Consider the problem of testing the simple null hypothesis $H:\theta =\theta_{0}$ against the one-sided alternative $K:\theta >\theta_{0}$. A significance level α test $\Phi_{0}$ with power function $\beta_{\Phi_{0}}\left(\theta \right)$ is said to be LMP if given any other level α test Φ with power function $\beta_{\Phi }\left(\theta \right)$, there exists Δ>0 such that $\beta_{\Phi_{0}}\left(\theta \right)\geq \beta_{\Phi }\left(\theta \right)$ for all θ∈K with $0\leq \theta -\theta_{0}\leq \Delta$.*

An LMP test is one of the best tests for detecting small deviations from null hypothesis, though it is not good at detecting all kinds of alternatives. In each stage of Algorithm 1, we aim at deciding whether the bandwidth is *k* or not, so small deviations in the coefficients of frequencies around *k* should be detected as possible as we can. Therefore, a LMP test is preferred in each stage. In Theorem 1, we will show that the Bayesian score test in each stage of Algorithm 1 is LMP in expectation.

**Theorem** **1.**
*Let $\bar{w}\left({\hat{f}}_{-k}\right)$ be the power function of the level α Bayesian score test of $H_{0}$ and let $w({\hat{f}}_{-k})$ be the power function of any level α test of $H_{0}$. The Bayesian score test of $H_{0}$ is LMP in expectation against all ${\hat{f}}_{-k}\in H_{A}$ with ${\hat{f}}_{-k}=\tau \xi$, where $0\leq \tau^{2}\leq \Delta$, that is $E_{\xi }\left[w(\tau \xi )\right]\leq E_{\xi }\left[\bar{w}\left(\tau \xi \right)\right]$.*


**Proof of Theorem 1.** Let $\bar{\beta }\left(\tau^{2}\right)$ be the power function of the level α score test of ${\bar{H}}_{0}$. Since ${\hat{f}}_{-k}=0$ and $\tau^{2}=0$ lead to the same distribution of y, the level α tests of $H_{0}$ and ${\bar{H}}_{0}$ lead to the same critical region. The same conclusion holds for any other level α tests of $H_{0}$ and ${\bar{H}}_{0}$ with power function $w({\hat{f}}_{-k})$ and $\beta (\tau^{2})$. It has been proved in Reference [25] that the one-side score test for one dimensional parameter is LMP. Therefore, for the given level α, there exists a Δ>0 such that for all $\tau^{2}\in {\bar{H}}_{A}$ with $0\leq \tau^{2}\leq \Delta$, the power function of score test for (Equation 10) is larger than that of any other test for (Equation 10)
(16)$$\begin{array}{c} \beta \left(\tau^{2}\right)\leq \bar{\beta }\left(\tau^{2}\right). \end{array}$$Let the critical region of the level α Bayesian score test be *A*, we have
(17)$$\begin{array}{cc} \beta (\tau^{2}) & =\int_{A}{\bar{L}}_{k}\left(\tau^{2};y\right)d\mu , \end{array}$$
where μ is a dominating measure. According to (Equation 9), ${\bar{L}}_{k}\left(\tau^{2};y\right)$ is the marginalized distribution of $L_{k}\left({\hat{f}}_{-k};y\right)$. According to (), every distribution shape of ${\hat{f}}_{-k}=\tau \xi$ with E(ξ)=0 and $E\left(\xi \xi^{T}\right)\propto \Sigma_{-k}$ leads to the same $S_{\Sigma_{-k}}$, and therefore the same power function. So (Equation 17) can be written as
(18)$$\begin{array}{c} \beta \left(\tau^{2}\right)=\int_{A}\left(\int L_{k}\left(\tau \xi ;y\right)\pi \left(\xi \right)d\xi \right)d\mu =\int \left(\int_{A}L_{k}\left(\tau \xi ;y\right)d\mu \right)\pi \left(\xi \right)d\xi =E_{\xi }\left[\bar{w}\left(\tau \xi \right)\right]. \end{array}$$According to (Equation 16) and (Equation 18), we have $E_{\xi }\left[w(\tau \xi )\right]\leq E_{\xi }\left[\bar{w}\left(\tau \xi \right)\right]$. □

To make Theorem 1 more intuitive, we give the following example.

*Example*: Suppose there are two alternatives to be tested in step *k*, $H_{A}^{l}={\hat{f}}_{-k}=\left[0.1,0,\cdots ,0\right]$, which has a small deviation 0.1 at a low frequency and $H_{A}^{h}={\hat{f}}_{-k}=\left[0,\cdots ,0.1,\cdots ,0\right]$, which has the same deviation at high frequency. The prior of ${\hat{f}}_{-k}$ is given as (Equation 8), which means the test pays more attention to the low frequency components. Then, the power of testing $H_{0}$ against $H_{A}^{l}$ will be larger than the power of testing $H_{0}$ against $H_{A}^{h}$ at the same significance level. As a result, the average power of the Bayesian score test is increased since there are more alternatives having small deviations at the low frequencies in a bandlimited GS.

Furthermore, the proof of Theorem 1 does not rely on the specific form of $\Sigma_{-k}$, which means that it will hold for different designs of $\Sigma_{-k}$. We can design different $\Sigma_{-k}$ to allocate attentions among frequency components to achieve various testing purposes with small sample size, for example, spectrum anomaly detection.

## 4. Numerical Analysis

### 4.1. Bandwidth Detection

An accurate bandwidth is helpful in various applications. For example, an accurate bandwidth in low-pass filter design for graph signals can help remove the noise and keep the original signal well; An accurate bandwidth can also help in choosing the minimal sample size in the sampling set design for graph signals. So in this section, we first validate the performance of bandwidth detection on an Erdos-Rènyi random graph with N=250 and the probability of edge presence being 0.25. The frequency coefficients of bandlimited GS are independently generated from a uniform distribution over the interval [0,1]. Gaussian noise is added to the GS to produce the observations, the signal to noise ratio (SNR) is calculated by $\mathrm{SNR}=10\log\left({∥f∥}_{2}^{2}/\left(N\sigma^{2}\right)\right)$. The performance of bandwidth detection is shown in bias of 1000 simulations, the SNR is set to 20 dB and vertices are sampled randomly in each simulation. In this simulation, we design two priors for the alternatives in our method follows the guideline in Section 3.2 and the significance level of the test is set to α=0.05. Prior 1 is given by (Equation 8) with δ=0.01. Prior 2 is given by
(19)$$\begin{array}{c} {\left(\Sigma_{-k}\right)}_{i,i}=\left\{\begin{array}{cc} 1-\frac{i-1}{M-k}+\delta & \;\mathrm{if}\;i\leq M-k+1;\\ \delta & \;\mathrm{if}\;i>M-k+1, \end{array}\right. \end{array}$$
with δ=0.01. To show the effectiveness of our prior designed for bandwidth detection, we compare the mean bandwidth obtained by giving ${\hat{f}}_{-k}$ our designed priors and uniform prior $\Sigma_{-k}=I_{\left(N-k)\times (N-k\right)}$, as shown in Figure 2. The test with uniform prior fails to detect the bandwidth when M=80, while the test with designed prior 1 and prior 2 performs well under the same sample size. When the sample size increases to 2N, the test with uniform prior turns out to have an acceptable performance, but still worse than the test with designed priors with M=80. This implies that the prior we designed for bandwidth detection is very helpful for saving the sample size. We can also find that the bandwidth detection performance of our method with prior 1 and prior 2 are comparable, which indicates the guideline of prior design in our algorithm improves the detection performance instead of a specific form of prior.

The performance of our method is also compared with the bandwidth estimation method in Reference [15], in which a dictionary of 80 bandlimited kernels is constructed with $\beta =10^{3}$ and the regularization parameter μ in it is set to 0.01 by cross validation. We first simulate the performance of bandwidth detection at different noise levels for bandwidth K={10,30,50}, the sample size is set to M=80. The results are shown in Figure 3. We can find that the bias and standard deviation (SD) of Algorithm 1 and sparsity based method [15] are similar for small bandwidth at low noise level. For large bandwidth, the bias and SD of our method are significantly smaller than those of Reference [15]. This is because the GS is no longer sparse for large bandwidth, which is out of the scope of sparsity based method [15]. At high noise level, Algorithm 1 is much more robust than Reference [15] especially when the bandwidth is close to the sample size.

In Figure 4, we show that the increase of sample size can help decrease the bias and SD of Algorithm 1. However, when the sample size is abundant, the performance improvement caused by the increase of sample size can be ignored.

### 4.2. Signal Estimation

Bandwidth is an useful prior when estimating signals from partial observations. In this section, we use the bandwidth detected by Algorithm 1 as a prior in signal estimation and a least squares (LS) estimator is used to estimate the signal with the form
(20)$$f^{\prime }=V_{K}{\left(V_{K}^{T}V_{K}\right)}^{-1}V_{K}^{T}y.$$

The estimation performance is evaluated by the normalized mean square error (NMSE), which is
(21)$$\mathrm{NMSE}=\frac{∥\left(f^{\prime }-f\right){∥}_{2}^{2}}{{∥f∥}_{2}^{2}}.$$

We first show the estimation performance on the same graph signal with that in Section 4.1. The bandwidth of signal varies from 10 to 60, the sample size is set to M=80 and the SNR is set to 20 dB. Our method is compared with two methods based on sparsity, namely BP [26] and BCS [21], which also select frequency components to estimate the signal. The result is show in Figure 5, we can find that as the bandwidth increases, the performance of the sparsity base methods degrade rapidly.

This is because the sparsity based methods select frequency components in the whole space and the sparsity constraint makes the amount of frequency components they selected as small as possible. Therefore, they are not fit for the application when the frequency spectrum is not sparse.

We also compare the performance of the signal estimation in real-data set with BP and BCS. The data set comprises 22 signals corresponding to the average temperature on January 1st in the intervals 1998–2019 measured by 119 stations in China [27]. Each station is identified with a vertex and the distance between 2 vertices is calculated by the haversine formula. A 3-NN unweighted graph is constructed. The GS is shown in Figure 6a and the corresponding spectrum is shown in Figure 6b.

i.i.d. Gaussian noise with SNR = 20 dB is added to the signal to generate observations. 200 simulations are implemented for each signal, the vertices are sampled randomly in each simulation. We can find that the temperatures distribute over the graph smoothly, so there are only a few low frequency components, which is sparse in spectrum. Since the GS is approximately bandlimited in real-data, the approximate bandwidth obtained by Algorithm 1 is used in (Equation 20). The average NMSE of all the simulations under different sample sizes is shown in Figure 7.

We can find that the estimation performance of our method is also better than the sparsity based methods in real-data. Since BP ensures the sparsity of the signal by $l_{1}$-norm, it results in obtaining the same signal values on each vertex in this experiment. BCS selects 14 nonadjacent frequency components in all the frequency components, including high frequencies and low frequencies. The approximate bandwidth obtained by Algorithm 1 is 12, which means we estimate the GS using the first 12 frequency components. We can find that if the GS is known to be bandlimited, the signal estimation performance can be improved by detecting the bandwidth accurately first and using the bandwidth as a prior in estimation. If the knowledge that the GS is bandlimited is not take advantage of, more samples are needed to achieve the same estimation performance.

## 5. Conclusions

In this letter, we proposed a multi-stage Bayesian score test approach for the bandlimited GS bandwidth detection with a small sample size. By customizing a prior for frequency coefficients being tested, we made the test more distinguishable from the models with bandwidths close to the assumed one. In practice, our method may obtain a bandwidth smaller than the true bandwidth *K* if there is a frequency band of which the coefficients are zeros in band, since we focus on testing the frequency coefficients close to the assumed one but ignore the others in the small sample size situation. We will try to improve the performance under this situation in the future. In addition, we would like to analyze how the different sampling sets affect the performance of bandwidth detection and design a sampling set to make the test more powerful.

## Figures and Tables

**Figure 1 sensors-21-00146-f001:**
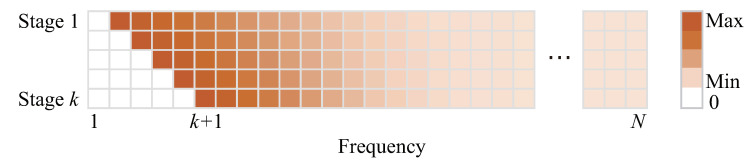
Illustration of the attentions on coefficients of different frequencies in different stages. Each line illustrates the attention distribution of one stage. Attention equals to 0 indicates that the corresponding frequency coefficient does not need to be tested.

**Figure 2 sensors-21-00146-f002:**
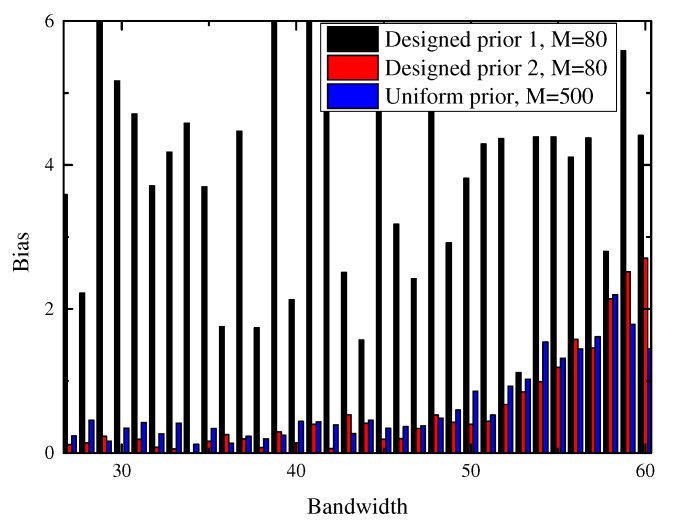
Bias of bandwidth detection under different priors.

**Figure 3 sensors-21-00146-f003:**
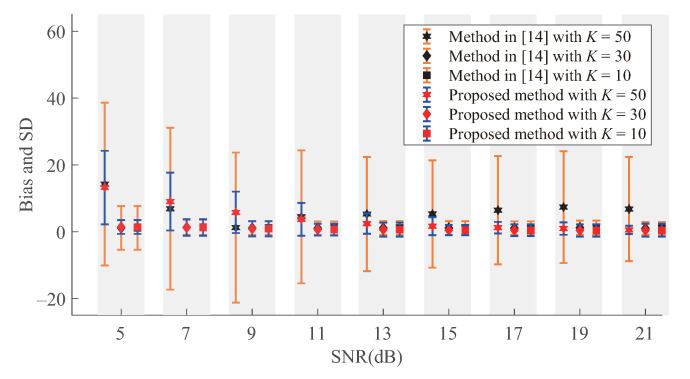
Bias and SD of bandwidth detection at different noise levels.

**Figure 4 sensors-21-00146-f004:**
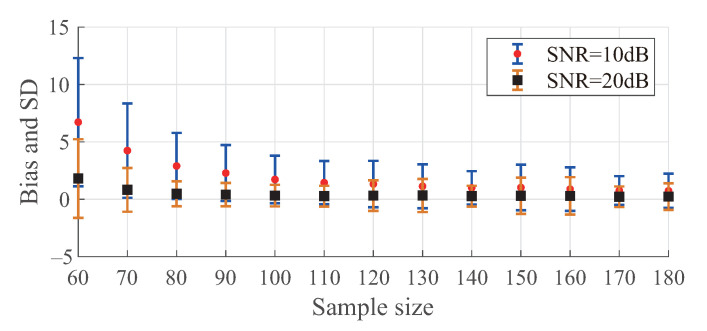
Bias and SD of bandwidth detection with different sample sizes when K=30.

**Figure 5 sensors-21-00146-f005:**
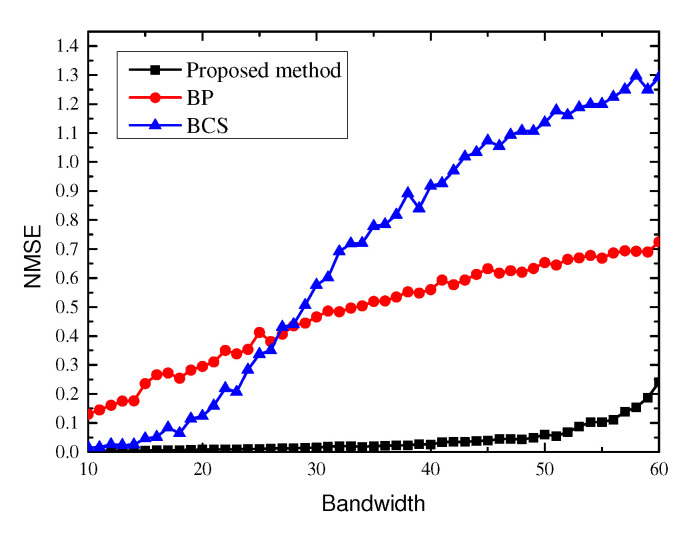
NMSE of graph signal (GS) estimation of different algorithms under different bandwidths.

**Figure 6 sensors-21-00146-f006:**
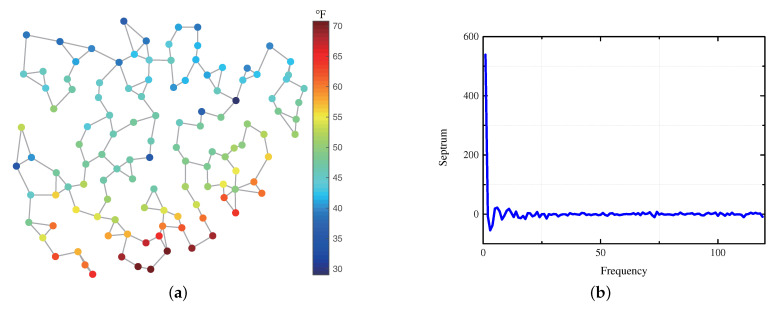
GS constructed from the temperature measured by 119 stations on 1st January 1998 and its spectrum. (**a**) GS on the 3-NN graph. (**b**) Spectrum of the GS in (**a**).

**Figure 7 sensors-21-00146-f007:**
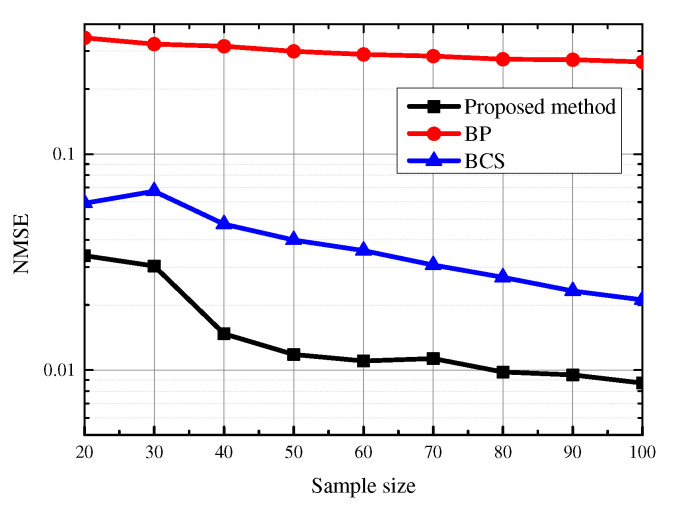
NMSE of GS estimation of different algorithms under different sample sizes.

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
