# Peer review of "Bandwidth Detection of Graph Signals with a Small Sample Size"

_sensors, 2020, doi:10.3390/s21010146_

Round 1
Reviewer 1 Report
Review of article: Bandwidth Detection of Graph Signals with a Small Sample Size
The authors deal with the detection of the bandwidth of bandlimited graph signal with a small sample size, but in my opinion this goal has not been successfully achieved. To increase some readability, I suggest the following improvements:
\item Throughout the paper, there are several inconsistencies in indexing. In lines~78-79, you define the bandwidth of GS. But in my opinion this definition is very strange if $K$ is equal to $N$. $\hat{f}$ is $N$-dimensional vector, do you know anything about $\hat{f}_k$ for $k>K=N$? Instead of "for all $k>K$", should be "for all $k\geq K$". Moreover, there is missing a definition of the bandwidth of $f$ as the smallest a such $K$ with the desired property, see [10].
\item Based on the bandwidth $K$ of $f$ in the previous paragraph: In lines~34-35, you write "If the bandwidth of a GS is $k$, only the first $k$ elements of frequency coefficients are non-zero." So, what is the bandwidth of $f$ ... $k$ or $K$? The potential reader must already be confused in this step. For the bandwidth $K$ of $f$, why must the first $K$ elements of frequency coefficients be non-zero? Why couldn't any of these $K$ values be zero?
\item In lines~69-70, why does the vertex set $\mathcal{V}$ and the edge set $\mathcal{E}$ have a different font style than the weighted adjacency matrix $\bf{ W}$?
\item Please make the paper self-contained. To do so, in lines~72-75, describe the spectral decomposition of graph Laplacian $\bf{L}$ more precisely. It seems that this requires only a few extra lines.
\item In lines~81-86, $k$ is any index or do you assume the bandwidth $k$ of $f$? Further, it took me some time to figure out, what $\Psi$ and $w$ mean; and I am still not sure, whether you and I have the same interpretation. The term i.i.d. Gaussian noise must be clarified, and throughout the paper, it is not mentioned in any way that it means~$\bf{I}$.
Due to the limited readability in Section~2 it was unfortunately not possible for me to observe the correctness of the described bandwidth detection in Section~3. The work is written in an unclear style. The paper is not suitable for publication in the journal Sensors with only minor revisions.
Reviewer 2 Report
- In my opinion, Bayesian compressive sensing (BCS) is one competing technique for the proposed detection method since BCS can simultaneously do signal reconstruction and learn the importance of the frequency components. More discussion and comparisons are needed.
- The choice of the prior on \hat{f}_{-k}, which I think is the main contribution of this work, however is heuristic. As a competitor, BCS is free of the prior selection.
- I don't quite follow the proof of Theorem 1. Can authors provide more details?
- The notations are a bit messy especially for the ones with \bar{} and \tilde{}.
Round 2
Reviewer 1 Report
Dear Authors, the manuscript "Bandwidth Detection of Graph Signals with a Small Sample Size". Despite multiple fixes, there are still some inconsistencies in your article. So, I suggest the following improvements:
\item The Sections 1 and 2 are now clearly introduced. In line 79, is there a mistake in indexing of the columns of $\bf{V}$?
\item In line~100, changing $\bf{\tilde{V}}$ to $\it{\bf{U}}$ have been a good choice. However, in lines~114-115, a different font style is used for $\bf{U}$.
\item In line~147, it would be appropriate to change "where the normalization factor $Z$ ..."
\item It would be appropriate to add an explanation of the term $\rm{tr}$ in the equation $(12)$. In particular, because the paper [19] is not easily available (for instance, I have no access to it).
\item In line~173, is the MLE result correct? In pdf format, I have some problem with its display.
The paper is suitable for publication in the journal Sensors after minor revisions.

Reviewer 2 Report
I am fine with the authors' responses.
Author Response
Thank you for your careful work. We appreciate your valuable comments that have helped improve this paper substantially.